# FoVAE: Reconstructive Foveation as a Self-Supervised Variational Inference Task for Visual Representation Learning

**Anonymous**                                                                            ANONYMOUS

**Anonymous**                                                                            ANONYMOUS

**Anonymous**                                                                            ANONYMOUS

## Abstract

We present the first steps toward a model of visual representation learning driven by a self-supervised reconstructive foveation mechanism. Tasked with looking at one visual patch at a time while reconstructing the current patch, predicting the next patch, and reconstructing the full image after a set number of timesteps, FoVAE learns to reconstruct images from the MNIST and Omniglot datasets, while inferring high-level priors about the whole image. In line with theories of Bayesian predictive coding in the brain and prior work on human foveation biases, the model combines bottom-up input processing with top-down learned priors to reconstruct its input, choosing foveation targets that balance local feature predictability with global information gain. FoVAE is able to transfer its priors and foveation policy across datasets to reconstruct samples from untrained datasets in a zero-shot transfer-learning setting. By showing that robust and domain-general policies of generative inference and action-based information gathering emerge from simple biologically-plausible inductive biases, this work paves the way for further exploration of the role of foveation in visual representation learning.

**Keywords:** foveation, reconstruction, variational autoencoder, predictive coding

## 1. Introduction

From shrimp to humans, the visual systems of many animal species have converged on *foveation*—a mechanism which allows an organism to attend to objects of interest by maintaining and orienting a small area of high acuity within the visual field (Land, 2011; Marshall et al., 2014). One advantage of this approach is its computational savings, as it would be prohibitively costly to cover the entire retina with high-density retinal cells (Akbas and Eckstein, 2017). We propose that animal foveation is not merely a convenient biological constraint, but is vital to visual representation building. Inspired by prior work on the human visual system, we aim to investigate the inductive biases that explicit foveation introduces in models of computer vision.

The human visual cortex is broadly composed of two streams—ventral and dorsal (Goodale and Milner, 1992). The ventral stream is involved in hierarchical form recognition and object representation. Meanwhile, the dorsal stream is thought to guide actions such as saccades and foveation, and represent positional information—largely in isolation from the ventral stream. Despite this separation, humans are biased toward foveating between distinct objects (Pajak and Nuthmann, 2013), indicating that the system of selecting the next foveation is based on an interaction between the "what" and the "where".

Psychophysical literature on human foveation reveals a bias towards looking at visual patches with high informational content (Rajashekar et al., 2007), and the fovea is primed for

the expected contents of the next fixation before the saccade begins (Kroell and Rolfs, 2022). These properties create a dynamic in which the visual system must select its next foveation to balance information gain with partial predictability. This view fits neatly with prevailing theories of predictive coding in the brain, including in the visual cortex (Rao and Ballard, 1999), which postulate that the brain learns about its environment via hierarchical Bayesian inference—predicting the environment from top-down priors while inferring information via bottom-up processing.

Convergent evidence from machine learning also attests to the power of learning by self-supervised environmental prediction (see Kojima et al., 2022; Cao et al., 2021; Oord et al., 2016). Furthermore, Pathak et al. (2017) demonstrates that a model can learn to explore video game environments with zero reward supervision by predicting its own representation of its next action and the resulting game state.

We aim to show that vision models can learn robust and general visual representations by performing next-element prediction in the context of a foveation sequence. To do so, we propose a hierarchical variational autoencoder (VAE) model that learns to look around an image in small patches across multiple time-steps, performing the following three objectives:

1. **Reconstruct the current patch** via variational inference.

2. **Generate the location of the next foveation, and predict the contents of the corresponding patch**, by predicting the top-level latent encoding of the next patch.

3. **After a set number of foveations, reconstruct the entire image** using the history of previously-seen top-level latent patch encodings.

Due to the competing losses for whole-image reconstruction and next-patch prediction, we expect the learned foveation patterns to balance information gain and predictability. That is, the contents of the next patch must be partially, but not wholly, captured in the model's priors, hopefully reflecting the same behaviors observed in humans (Rajashekar et al., 2007; Kroell and Rolfs, 2022). Finally, we hypothesize that by decoupling the patch representation and foveation mechanism from the reconstruction of the whole image, we introduce a crucial inductive bias for representing patches and the relations between them, as opposed to shallow statistical correlations between pixels. This may allow the foveation and reconstruction policies learned by the network to generalize to unseen datasets.

## 2. Background

Most state-of-the-art approaches to computer vision tasks rely on convolutional neural networks (CNNs). These models process every patch of the image simultaneously, and are conceptually equivalent to a retina uniformly covered with cells of equal resolution. CNNs are highly inaccurate models of human vision, and fail to account for a wide range of human behaviors (Bowers et al., 2022; German and Jacobs, 2020; Brendel and Bethge, 2019). Although soft attention methods (Xu and Saenko, 2016; Sharma and Jalal, 2021; Song et al., 2022) improve upon the performance of CNNs for downstream tasks, the attentional patterns learned bear little resemblance to human foveation patterns (Jain and Wallace, 2019; van Dyck et al., 2021). Some work has shown the benefits of explicitly modeling

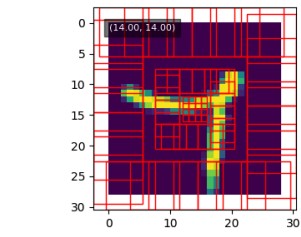
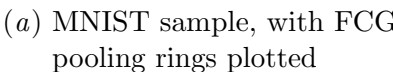

(*a*) MNIST sample, with FCG pooling rings plotted

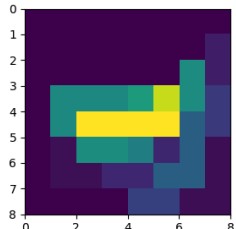

(*b*) Resulting foveated patch, after applying FCG

Figure 1: Example of Foveal Cartesian Geometry (FCG) applied at a centered position in an MNIST sample. White space in (a) is padded with the value of the closest non-padding pixel. Coordinates in the top left denote the the center of the patch.

foveation, achieving equivalent or better performance on image-based tasks at a fraction of the computational cost (see Pramod et al., 2022; Lukanov et al., 2021).

In the last decade, VAEs (Kingma and Welling, 2014; Higgins et al., 2016) have emerged as one of the leading approaches to self-supervised representation learning. Recently, Child (2021) have shown that a very deep hierarchical VAE (VDVAE) is able to achieve very high likelihood on natural image modeling. Recurrent VAE models have previously been used for image reconstruction, such as DRAW (Gregor et al., 2015), AIR (Eslami et al., 2016) and DooD (Liang et al., 2022). Most similarly to the current work, AIR and its extension SQAIR (Kosiorek et al., 2018) learn to represent visual scenes in an unsupervised manner by variationally reconstructing scene objects in discrete time-steps. At test time, AIR generalizes to unseen numbers of objects, novel object orientations, and represents complex objects in part-whole hierarchies.

## 3. Method

The proposed model (FoVAE) learns to represent image inputs one patch at a time via a variational reconstructive task. See Appendix A for full details of the model operation.

### 3.1. Foveal patches

Departing from prior VAE scene representation methods (Eslami et al., 2016; Nash et al., 2017), FoVAE is applied to small input patches without explicitly assuming the existence of objects. This allows the model to tractably build representations directly from input pixels without requiring CNN features or looking at the entire image. For a given foveal center in image coordinates $a = \{x, y\}$, a foveated representation of the image is computed using Foveal Cartesian Geometry (Lukanov et al., 2021; Martinez-Carranza and Altamirano Robles, 2006). As shown in Figure 1, pixels in a small radius around the center of the foveation are sampled directly, representing the high-resolution area of the biological fovea. The remainder

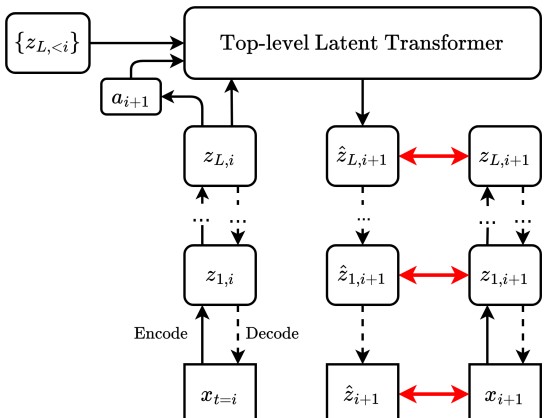

Figure 2: Simplified model schematic. At timestep $t = i$, the model infers the top-level hierarchical latent vector $z_{L,i}$, infers the next foveation position $a_{i+1}$ and generates the top-level latent $\hat{z}_{L,i+1}$ of the next patch at location $a_{i+1}$. After foveation to $a_{i+1}$, loss is computed as the likelihood of the observed next-step latent vectors given the ones generated from the previous step $p(z_{k,i+1}|\hat{z}_{k,i+1})$. Likelihood loss is denoted by the red arrows. Image reconstruction and details of LVAE are omitted.

of the input patch is filled by averaging pixel values of farther-away pixels, with the radius of the average pooling proportional to the pixels' distance from the center[1].

### 3.2. Foveation

As shown in Figure 2, at each timestep $t = i$ FoVAE first reconstructs the current foveal patch through variational inference. Following Child (2021), the proposed architecture is a Deep Ladder VAE (LVAE; Sønderby et al. (2016)), consisting of a deterministic bottom-up feature extraction network $D$, which conditions, through inference network $Q$, a hierarchical top-down generative network $P$. This yields a set of hierarchical latent vectors $z_{1 \leq k \leq L, t=i}$, where $L$ is the number of hierarchical layers of the VAE.

At timestep $i$, top-level latent vectors $z_{L,t<i}$ of all patches seen so far condition the distribution of the location of the next foveation $a_{t=i+1}$. After sampling the next location, we predict the top-level latent vector of the *next patch*, conditioned on the top-level latent vectors of the patches at all prior timesteps and the location of the next foveation. In line with the two-stream hypothesis of human vision, the networks which predict the location and content of the next foveation share no parameters.

After sampling the next top-level latent vector $\hat{z}_{L,i+1}$, we predict the contents of the next patch with top-down generative inference, using the same network $P$ as for current-patch generation. After next-patch prediction, foveation to the next location $a_{i+1}$ is performed.

---

1. In practice this is implemented as several concentric square rings, with farther rings corresponding to larger radii of average pooling across neighboring pixels. We omit the foveal patch generation algorithm for brevity, please see Martinez-Carranza and Altamirano Robles (2006) for more details.

Table 1: Average whole-image reconstruction losses for networks trained on (a) MNIST and (b) Omniglot, on the validation sets of both datasets.

| (a) | | | | (b) | | |
|---|---|---|---|---|---|---|
| | MNIST | $\rightarrow$ Omniglot | | | Omniglot | $\rightarrow$ MNIST |
| Base | 7.8 | 24.5 | Base | | 8.7 | 40.8 |
| + *mask-to-last* | **6.9** | **17.5** | + *mask-to-last* | | **6.9** | **21.5** |
| + *big-patch* | - | - | + *big-patch* | | 8.2 | 25.1 |

The reconstruction loss for this component is the log probability of the real latent vectors for the next patch given their predictions, at all layers.

After the foveation to the next patch, the processing of the new current patch proceeds as described above. Finally, after $T$ timesteps of foveation, the entire input image is reconstructed from the set of observed top-level foveal latent vectors $z_{L,t \leq T}$. This action is performed using the same next-patch prediction process, while forcing the location of the next patch $a$. The final loss objective of FoVAE is the sum of the weighted reconstruction losses for: current patch reconstruction, next patch prediction, and whole-patch reconstruction, along with the corresponding KL divergence terms, for all foveation timesteps[2].

## 4. Experiments

We first evaluate FoVAE on the MNIST (Lecun et al., 1998) and Omniglot (Lake et al., 2015) datasets in isolation. Our metric is whole-image reconstruction loss, as it comprises the entire image and achieving a low value of this loss requires learning good patch representations, aggregation of patch information across timesteps, and a sensible foveation strategy. Whole-image reconstruction loss also serves as a proxy for the computationally-expensive image log likelihood calculations. In addition, we evaluate whole-image reconstruction loss on Omniglot for networks trained on MNIST, and vice versa. In order to do well on this out-of-domain task, the network must learn patch representations that generalize across domains. Finally, we also analyze the learned patterns of foveation and the quality of the reconstructions.

We evaluate FoVAEs with two latent layers, with $z_1$ having 20 latent dimensions and $z_2$ having 10. This accounts for more featural variation in the low-level details than in higher-level patch descriptions[3]. As a starting point, each foveal patch is $6 \times 6$ pixels, with the middle $4 \times 4$ being the fovea, and the periphery being located close to the center (see top row of Figure 3). In the `big-patch` variant, we explore expanding the periphery to two rings with a larger degree of pooling on the second ring. Finally in the `mask-to-last` variant we explore restricting the next patch prediction network to only condition $z_{L,i+1}$ on the top-level vector of the current step $z_{L,i}$.

As shown in Tables 1(a) and 1(b), whole-image reconstruction loss increases as expected when transferring across domains. We also observe the following phenomena:

---

2. Please see Appendix A for the full loss formulation.
3. Please see Appendix B for further hyperparameters.

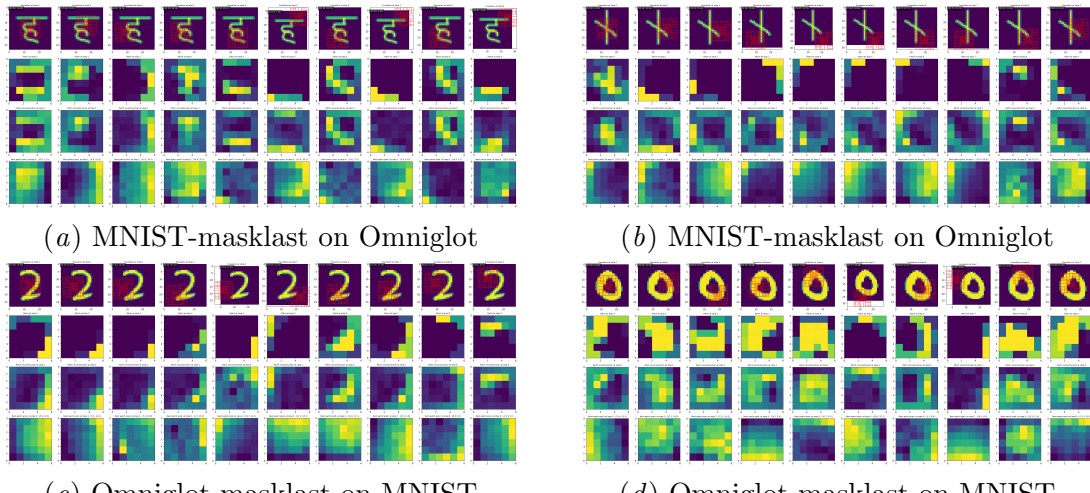

(a) MNIST-masklast on Omniglot     (b) MNIST-masklast on Omniglot

(c) Omniglot-masklast on MNIST     (d) Omniglot-masklast on MNIST

Figure 3: Learned foveation behavior of: (a,b) MNIST-masklast model on two Omniglot samples and (c,d) Omniglot-masklast model on two MNIST samples. Top row: movement of the foveal sampling grid. Second row: resulting foveal patch. Third row: variational reconstruction of patch. Fourth row: next-patch prediction. Each column is one timestep. All foveations start in the center of the image.

**1. MNIST transfer to Omniglot achieves, on average, better performance than transfer from Omniglot to MNIST.** This result is unexpected, because MNIST is a simpler dataset, and thus we would expect the priors learned from MNIST to skew more strongly toward digit-like shapes (which indeed it does, see qualitative analysis below). It is possible that this result stems from the fact that MNIST is simpler to learn to reconstruct to saturation. In contrast, Omniglot is more difficult and varied than MNIST, and the network is forced to rely on shallow strategies such as statistical correlation, which do not generalise. Future work will investigate whether this disparity persists if the networks are trained to saturation on their respective datasets.

**2. Conditioning only on the current top-level latent vector is beneficial** both in- and out-of-domain, on both datasets. We believe that this result stems from a fundamental difference in the behaviors learned by the model, due to the change in inductive biases between the two conditions. Removing historical foveation information may force the next patch prediction network to pay more attention to local features such as lines, edges and peripheral cues. This guides the next foveation to areas of high *local* predictability while still collecting *globally*-novel information and avoiding the drastic changes in foveation position, as far-away patches are usually not predictable. Meanwhile, access to historical foveations may allow the network to have stronger global priors about the contents of every image patch. This would allow it to optimize more strongly for predictability over information gain, reducing its performance.

**3. Adding more global context in the form of a larger periphery to the foveal patch reduces performance**. As above, this finding is also likely due to the change in the

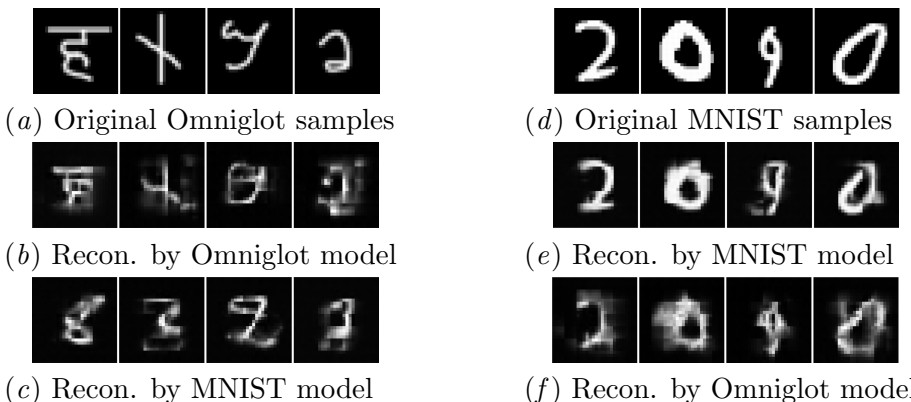

(a) Original Omniglot samples      (d) Original MNIST samples

(b) Recon. by Omniglot model      (e) Recon. by MNIST model

(c) Recon. by MNIST model      (f) Recon. by Omniglot model

Figure 4: In- and out-of-domain reconstructions of Omniglot and MNIST validation samples.

balance of surface-level priors-based global predictability, as opposed to incentivizing local exploration in the absence of global information.

Our quantitative findings are supported by visual analysis of foveation patterns and sample reconstruction quality. Figures 3(a) and 3(b) shows the foveation, patch reconstruction, and next-patch prediction process for a FoVAE model trained on MNIST and tested on Omniglot samples. Foveations largely follow the contours of the figures, maximizing local predictability. Individual patch reconstructions are reasonable despite many of the contours of the figure not matching those within the domain of MNIST. However in Figure 3(b), next-patch prediction struggles due to the dissimilarity of the sample to the training set, which is reflected in the whole-image reconstruction in the second column of Figure 4(c) [4].

Figures 3(c) and 3(d) and Figure 4(f) show an analogous set of visualizations for FoVAE trained on Omniglot and tested on MNIST. In Figure 3(c), FoVAE fails to follow the contour of the novel sample, with a correspondingly poor reconstruction. In Figure 3(d) it fares better, successfully foveating around the shape. Both samples exhibit subpar next-patch prediction, relying on broad directional predictions rather than predicting specific features. As mentioned above, this may be due to the Omniglot-specific biases not yet refined into domain-general behavior due to the complexity of the Omniglot dataset. However, due to the variety of patches presented in Omniglot, the reconstructions of MNIST by the Omniglot are overall more faithful to the MNIST domain (Figure 4(f)), while the influence of MNIST-specific priors on the reconstructions of Omniglot in Figure 4(c) is evident.

## 5. Conclusions

In this paper, we present the first steps toward a more biologically-plausible model of visual representation learning, driven by a self-supervised reconstructive foveation mechanism. Tasked with looking at one patch at a time while reconstructing the current patch, predicting the next patch, and reconstructing the full image after a set number of timesteps, FoVAE

---

4. Future work would explore continuous learning in this setting, where as in neural predictive coding, large errors in next-patch prediction would drive domain-adaptive behavioral shifts.

learns to attend to object contours in the MNIST and Omniglot datasets, while inferring high-level priors about the whole image. In line with the two-stream hypothesis of the human visual system, theories of Bayesian predictive coding in the brain and prior work on human foveation biases, FoVAE combines bottom-up input processing with top-down learned priors to reconstruct its input, choosing foveation targets that balance local feature predictability with global information gain. We further demonstrate a surprising result, in which constraining the amount of information the model has about the content of its historical foveations actually improves reconstructive performance. FoVAE is able to transfer its priors and foveation policy across datasets, and is able to integrate novel bottom-up inputs from MNIST with previously-learned top-down priors from Omniglot to reconstruct samples from the new dataset in a zero-shot transfer-learning setting. By showing that robust and domain-general policies of generative inference and action-based information gathering emerge from simple biologically-plausible inductive biases, this work paves the way for further exploration of the role of foveation in visual representation learning.

## 6. Limitations and Future Work

Many aspects of the presented model have been left unexplored. Firstly, we evaluate FoVAE on two simple datasets, which are sufficient to show the generalization capabilities of the model, but are too simple to require truly robust foveation policies. The simplicity and small size of the inputs also do not evidence the computational benefits of the foveation mechanism over methods that process the entire input at once—indeed, the presented model would scale effortlessly to $1024 \times 1024$ pixel images at near-constant memory use, which would strain most CNN architectures and completely rule out the use of MLP-based prior approaches like DRAW and AIR. Additionally, using more complex datasets such as the single-object ImageNet (Deng et al., 2009) and multi-object GQA (Hudson and Manning, 2019) would allow us to explore whether the learned foveation patterns of the model exhibit the same part-whole hierarchy bias (Biederman, 1995) and object-centered bias (Pajak and Nuthmann, 2013) present in human foveation and object recognition.

Furthermore, we only present models with two variational inference layers, whereas the results of Child (2021) required a VAE architecture with close to 80 layers. This limitation is due to the well-attested difficulty of getting deep VAEs to converge stably (Dehaene and Brossard, 2021). The depth of the VAE and the choice of which layer(s) to use to infer the next foveation position, directly affect the breadth of the priors modeled by the network. We theorize that if the VAE is made very deep, with the next foveation position conditioned not on the top-level latent $z_L$ but on latents derived from layer $i$ in the middle of the layer stack, the layers above $i$ will start to encode priors about input patches which are position-invariant. The emergence of spatial and other invariants at increasingly higher layers of the hierarchy can be studied by traversing the latent space of high-level priors, manipulating the prior and observing the changes in the generated image. Studying representations of the same patch in the contexts of different images would likely also be illuminating—for example, the representation of the circles present in the digits 9 and 8. An invariant coding of the two at a particular level of the hierarchy would indicate that the model has learned to represent the same "part" separately from the context of the larger "whole", with the larger context being delegated to higher levels in the hierarchy in order to select the next foveation.

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

# Appendix A. Operation details

## A.1. Current patch reconstruction

Firstly, at each timestep the FoVAE model reconstructs the current foveal patch through variational inference. Following Child (2021), the proposed architecture is a Deep Ladder VAE (LVAE; Sønderby et al. (2016)), consisting of a deterministic bottom-up feature

extraction network $D$ which conditions, through inference network $Q$, a hierarchical top-down generative network $P$. For an input patch $x$, at each layer of the hierarchical model $1 < n < L$:

$$d_n = D_n(d_{n-1}) \tag{1}$$

$$\{\hat{\mu}_{q,n}, \hat{\sigma}_{q,n}\} = Q_n(d_n) \tag{2}$$

$$\{\mu_{p,n}, \sigma_{p,n}\} = P_n(z_{n+1}) \tag{3}$$

where $d_0 = x$, and for the top layer, $\mu_{p,L} = \hat{\mu}_{q,L}$ and $\sigma_{p,L} = \hat{\sigma}_{q,L}$.

The top-down and bottom-up parameters are then combined through precision-weighted averaging, producing the final parameters of the latent posterior distribution:

$$\sigma_{q,n} = \frac{1}{\hat{\sigma}_{q,n}^{-2} + \sigma_{p,n}^{-2}} \tag{4}$$

$$\mu_{q,n} = \frac{\hat{\mu}_{q,n}\hat{\sigma}_{q,n}^{-2} + \mu_{p,n}\sigma_{p,n}^{-2}}{\hat{\sigma}_{q,n}^{-2} + \sigma_{p,n}^{-2}} \tag{5}$$

Combining the parameters via averaging (as in Sønderby et al. (2016)) as opposed to parametrically (as in Child (2021)) ensures that the top-down and bottom-up parameters lie in the same space and allows us to avoid training a separate network for prior-only generation.

Finally, the $n$-th layer latent vector for the current patch $z_n$ is sampled from a normal distribution parameterized by $\mu_{q,n}$ and $\sigma_{q,n}^2$:

$$z_n \sim \mathcal{N}(\mu_{q,n}, \sigma_{q,n}^2) \tag{6}$$

At the end of the generative process, $z_0$ is taken to be a variational reconstruction of the original input $x$. The Gaussian log likelihood of the input forms the first component of the model's loss:

$$\mathcal{L}_{cr} = \log p(x|z_0) \tag{7}$$

In line with other VAE approaches, we also constrain the KL divergence of each latent vector from a standard-normal Gaussian prior:

$$\mathcal{L}_{cd} = \sum_{n=1}^{L} KL(z_n \parallel \mathcal{N}(0,1)) \tag{8}$$

### A.2. Inferring the next foveation

At timestep $i$, top-level latent vectors of all patches $z_L$ seen so far condition the distribution of the location of the next foveation $a_{i+1}$:

$$\{\mu_{a,i+1}, \sigma_{a,i+1}\} = NPP_a(\{z_{L,1}, z_{L,2} \ldots z_{L,i}\}) \tag{9}$$

$$a_{i+1} \sim \mathcal{N}(\mu_{a,i+1}, \sigma_{a,i+1}^2) \tag{10}$$

where $\mu_a, \sigma_a, a$ are 2-dimensional vectors describing the mean, standard deviation and sampled values of the $x$ and $y$ locations of the next foveation, respectively. After sampling, $a_{i+1}$ is normalized to lie in $[-1, 1]$ by a scaled sigmoid function.

After sampling the next location, we predict the top-level latent vector of the *next patch*, conditioned on the top-level latent vectors of the patches at all prior timesteps and the location of the next foveation.

$$\{\hat{\mu}_{p,L,i+1}, \hat{\sigma}_{p,L,i+1}\} = NPP_p([z_{L,1} \ldots z_{L,i}], a_{i+1}) \tag{11}$$

$$\hat{z}_{L,i+1} \sim \mathcal{N}(\hat{\mu}_{p,L,i+1}, \hat{\sigma}_{p,L,i+1}^2) \tag{12}$$

The next-patch prediction networks $NPP_a$ and $NPP_p$ are implemented as Vision Transformers (ViT, Dosovitskiy et al. (2020)), which excels at modeling long-range dependencies between elements of arbitrary-length sequences. Both networks take as input the top-level latent vectors $z_L$ seen so far, and for $NPP_p$, the 2-dimensional coordinates of the center of the next foveation $a_{i+1}$ are appended to each input latent vector. Similar to autoregressive teacher forcing approaches (Williams and Zipser, 1989; Radford et al., 2018), the *NPP* networks infer the parameters of the next top-level latent vector in the input sequence, and the true top-level latent vector of the next patch is added to the input sequence at the next timestep. In line with the two-stream hypothesis of human vision (ventral and dorsal, as in Goodale and Milner (1992)), $NPP_a$ and $NPP_p$ share no parameters.

After sampling the next top-level latent vector $\hat{z}_{L,i+1}$, we predict the contents of the next patch with top-down generative inference, using the same network $P$ as for current-patch generation (Equation 3):

$$\{\hat{\mu}_{p,n,i+1}, \hat{\sigma}_{p,n,i+1}\} = P_n(\hat{z}_{n+1,i+1}) \tag{13}$$

$$\hat{z}_{n,i+1} \sim \mathcal{N}(\hat{\mu}_{p,n,i+1}, \hat{\sigma}_{p,n,i+1}^2) \tag{14}$$

$\hat{x}_{i+1} = \hat{z}_{0,i+1}$ is the prediction of the contents of the next patch. After next-patch prediction, foveation to the next location $a_{i+1}$ is performed. The reconstruction loss for this component of the model is the log probability of the real latent vectors for the next patch given their predictions, at all layers.

$$\mathcal{L}_{nr} = \log p(x_{i+1}|\hat{x}_{i+1}) + \sum_{n=1}^{L} p(z_{n,i+1}|\hat{z}_{n,i+1}) \tag{15}$$

The KL divergence losses are calculated both on the divergence of the latent vectors from a standard-normal prior, and on the divergence of the next position from a standard-normal prior[5].

$$\mathcal{L}_{nd} = \sum_{n=1}^{L} KL(\hat{z}_{n,i+1} \parallel \mathcal{N}(0, 1)) \tag{16}$$

$$\mathcal{L}_{ad} = KL(a_{i+1} \parallel \mathcal{N}(0, 1)) \tag{17}$$

After the foveation to the next patch, the processing of the new current patch proceeds according to Section A.1.

---

5. The latter term is appropriate as positions in both $x$ and $y$ dimensions are normalized to lie in $[-1, 1]$ and this constraint encourages the network to foveate to all locations equally *a priori*.

### A.3. Whole input reconstruction

Finally, after $T$ timesteps of foveation, the entire input image is reconstructed from the set of foveal latent vectors $\{z_{L,i}\}, 1 \leq i \leq T$. This is performed using the same next-patch prediction process in Section A.2, while forcing the location of the next patch $a$, instead of sampling it from the appropriate distribution[6].

$$\mathcal{L}_{ir} = \sum_a \log p(x_{T+1}|\hat{x}_{T+1}) \tag{18}$$

This objective performs the same role as the "canvas painting" objective of DRAW (Gregor et al., 2015) and AIR (Eslami et al., 2016), while avoiding the computational cost of a potentially-huge canvas.

### A.4. Optimization

The final loss objective of the model is a summation of the reconstruction losses from the preceding sections—current patch reconstruction, next patch prediction, and whole-patch reconstruction—and the corresponding KL divergence terms, for all foveation timesteps. As in $\beta$-VAE (Higgins et al., 2016), each term is weighted by a $\beta$ parameter.

$$\mathcal{L} = \sum_{i=1}^{T} \beta_{cr}\mathcal{L}_{cr} + \beta_{nr}\mathcal{L}_{nr} + \beta_{ir}\mathcal{L}_{ir} + \beta_{cd}\mathcal{L}_{cd} + \beta_{ad}\mathcal{L}_{ad} + \beta_{nd}\mathcal{L}_{nd} \tag{19}$$

This objective is optimized jointly, applying the reparameterization trick (Kingma and Welling, 2014) when sampling from normal distributions is required to allow backpropagation.

## Appendix B. Model hyperparameters

All mappings of $D$, $Q$ and $P$ are implemented by feed-forward neural networks with two layers of 512 neurons each, each layer preceded by a GELU nonlinearity (Hendrycks and Gimpel, 2020). Thus, each $d_n \in \mathbb{R}^{512}$. Each network performs 10 steps of free foveation, after which it is used to reconstruct the entire input image using foveation to forced locations. Each network is trained using the Adam optimizer on its respective dataset for 5000 epochs, using a learning rate of 0.0001.

---

6. In practice we only reconstruct a randomly-sampled 10% of possible input image patches, due to significant computational savings with no decrease in performance.

