# OpenReview forum: "FoVAE: Reconstructive Foveation as a Self-Supervised Variational Inference Task for Visual Representation Learning"
_NeurIPS.cc/2023/Workshop/Gaze_Meets_ML — Gaze Meets ML 2023 Poster_

### Official Review · Reviewer_LRXu · 2023-10-21
**FoVAE  for Visual Representation Learning**

**Rating:** 5
**Confidence:** 3

**Review:**

The paper proposes a novel method (FoVAE) for Visual Representation Learning, based on the biologically-plausible concepts of human foveation biases and theories of Bayesian predictive coding, as long as ventral/dorsal human visual stream hypothesis. FoVAE  learns to represent the input image via a variational reconstruction task, performed firstly on a sequence of foveated patches and lastly on the whole input image. Patch foveation is performed by Foveal Cartesian Geometry, while the foveation locations are sampled from a distribution that is conditioned by top-level latent vectors of a subset of the patches seen so far. While the model seem theoretically well-founded, the work is preliminary and needs further evolution, especially from the experimental point of view.  The paper is well written and the proposed method is original and novel.

Pros:
+ The model is theoretically well-grounded and aims at a biologically-plausable representation learning.
+ The (very preliminary) results seem promising, given that the model is capable to achieve the reconstruction of toy tasks.

Cons:
- The "Method" section of the paper (and Figure 2) gives an extremely high-level view of the proposed model,  that hinders the comprehension of the details in the model pipeline. Even if the appendix sheds some light on the inner workings, the paper should suffice in giving a better comprehension, and this is not the case.
- The experimental findings are very preliminary, given that the model has been tested solely on extremely simple datasets (as ackowledged by the authors in the Limitation section). The authors claim to tackle an out-of-domain task when  transferring from MNIST to Omniglot, while the two datasets share very similar visual appearances.
- The main paper message is the ability to achieve Visual Representation Learning. The best way to prove the goodness of the learned representation should be to use it for a downstream task (e.g. classification) that differs from the reconstruction one.
- The computational/time complexity of the proposed approach has not been investigated.

---

### Official Review · Reviewer_BJMx · 2023-10-22
**Novel VAE-based architecture for foveated fixations on images, with limited set of experiments**

**Rating:** 8
**Confidence:** 3

**Review:**

The authors present a novel VAE-based architecture for foveation and attention. For an initial fixation at the center, it generates a latent and reconstructs the foveated patch as well as predicts the next foveation point and patch. A sequence of such fixations is generated and the used to predict the full image. This aligns with human vision, as it has a predefined expectation of what it sees at the next foveation point.

Strengths:
* Present a novel, multifaceted architecture to model human foveation centered around a sequence of latent representations of the scene. Human vision is modelled by requiring the architecture to reconstruct the current and next foveation patches as well as the full image.
* Compare next foveation point prediction using both the full sequence of latents as well as only the last latent, with the latter clearly outperforming the former.
* Evaluate generalization across the two datasets MNIST and Omniglot in terms of image reconstruction quality, with a marked decrease in performance observed (but less when transferring from MNIST to Omniglot).

Weaknesses:
* A comparison of the generated foveation points with known human eye tracking data would be beneficial. None I am aware of exist for MNIST, but for other datasets eye tracking data is readily available and may be used to determine how "human-like" the proposed model actually behaves.
* As mentioned by the authors themselves, the two datasets used (MNIST and Omniglot) are not particularly representative. In particular, both are black-and-white handwritten character datasets.
* In section 3.2 the text suggests that the full sequence of latents is used to predict the next foveation point ("At timestep i, ..."), but the figure shows only the last latent. Please describe both variants and clearly state what is shown in the figure itself.

---

### Meta-Review · Area_Chair_JkA2 · 2023-10-26

**Recommendation:** Accept (Poster)
**Confidence:** 5

**Metareview:**

The authors present a biologically-inspired visual attention deep learning method. The paper is well written and well referenced. Reviewers have provided very meaningful comments. While evaluation of the proposed methodology on datasets containing eye gaze is much desirable, in the current stage I would expect the authors to address 'less tedious' tasks such as improving figure 2. Providing performance information such as computational/time complexity and classification in downstream task would also be welcomed.

---

### Decision · Program_Chairs · 2023-10-26

Accept (Poster)